# Assessment of Diabetes-Related Knowledge and Dietary Patterns Among Type 2 Diabetes Mellitus Patients in Central Saudi Arabia: Insights for Tailored Health Education Strategies

**DOI:** 10.3390/healthcare13030233

**Published:** 2025-01-24

**Authors:** Aseel Awad Alsaidan, Mohammed Ibrahim Alanazi, Ashokkumar Thirunavukkarasu

**Affiliations:** Department of Family and Community Medicine, College of Medicine, Jouf University, Sakaka 72388, Aljouf, Saudi Arabiaashokkumar@ju.edu.sa (A.T.)

**Keywords:** type 2 diabetes, diet, duration, knowledge, correlation, Saudi Arabia

## Abstract

**Background and Objectives**: Understanding the knowledge and dietary patterns of type 2 diabetes mellitus (T2DM) patients is essential to identify gaps and design tailored health education strategies to improve self-management and clinical outcomes. We assessed the diabetes-related dietary patterns, knowledge, and associated factors of T2DM patients. **Methods**: The study utilized a cross-sectional design, surveying 363 T2DM patients using a validated and pretested questionnaire. Knowledge levels were categorized as low (<50%), medium (50–75%), and high (>75%), and dietary patterns were classified as unhealthy (<34%), moderately healthy (34–67%), and healthy (>67%). We analyzed the data using the Statistical Package for the Social Sciences (SPSS, version 23.0). The authors tested the relationship between diabetes-related dietary patterns and knowledge with Spearman’s analysis. The multivariate regression approach established the factors associated with these two components. **Results**: The findings revealed that 36.4% of participants had low knowledge, 34.4% had medium knowledge, and only 29.2% demonstrated high knowledge. Regarding dietary patterns, 34.4% were categorized as unhealthy, 33.3% as moderately healthy, and 32.3% as healthy. The authors found that these two domains were positively correlated (rho = 0.649, *p* = 0.001). Diabetes-related knowledge was significantly associated with gender (*p* = 0.018), duration of diabetes (*p* = 0.001), and patients with a family history of T2DM (*p* = 0.007). The dietary pattern was significantly associated with age (*p* = 0.001), duration of diabetes (*p* = 0.032), and presence of other chronic diseases (*p* = 0.028). **Conclusions**: The findings underscore the need for targeted health education strategies that address gaps in dietary knowledge and promote healthier eating behaviors among T2DM patients in Central Saudi Arabia.

## 1. Introduction

Diabetes mellitus is one of the twenty-first century’s major health concerns, impacting about 537 million people around the world in 2021 and is expected to reach 783 million by 2045 [1]. Type 2 diabetes mellites (T2DM), which is a chronic metabolic disease, is the most prevalent kind of diabetes and is characterized by abnormalities in insulin production and action, either of which may predominate [2,3]. T2DM is caused by several significant risk factors, including poor dietary intake, decreased physical activity, obesity, advancing age, a family history of diabetes, ethnicity, and changes in lifestyle [3]. The Gulf Cooperation Council (GCC) countries, including Saudi Arabia (KSA), face a diabetes crisis, with some of the highest prevalence rates globally. Rapid urbanization, reduced physical activity, and a shift toward processed food consumption have significantly contributed to the rising burden of Type 2 Diabetes Mellitus (T2DM) in the region [4,5]. Similarly to other GCC countries, the prevalence rate of T2DM is alarming and widely differs across different regions in the KSA due to variations in lifestyle factors, genetic predispositions, urbanization levels, and access to healthcare [4,6].

Diabetes-related dietary knowledge encompasses ideas and procedures pertaining to diet and health, diet and illness, food value and its explanation of nutrients, and guidelines that must be adhered to. Furthermore, knowledge of dietary recommendations provided by healthcare professionals influences a patient’s choice of foods and eating habits, helping them make informed decisions to manage their diabetes effectively [7,8]. However, globally and in the KSA, several studies observed critical gaps in knowledge in these aspects. For instance, a survey in Thailand by Phoosuwan et al. highlighted poor knowledge among T2DM patients, with education, employment, and the presence of complications being significant associated factors [9]. Similarly, Ferreira et al. explored diabetes knowledge in Portugal and observed significant variability across different domains of knowledge, influenced by factors such as age, education, and settings [10].

In the context of Saudi Arabia, a study conducted in Madinah reported that a sizable proportion of T2DM patients had average knowledge related to diabetes, with responses varying across different knowledge domains. These findings highlight regional and global gaps in diabetes-related knowledge that necessitate targeted educational interventions [11,12]. Similarly to diabetes-related knowledge, dietary patterns and adherence to recommended dietary patterns are also crucial in glycemic control among T2DM patients [13,14]. Several studies observed poor dietary patterns in GCC countries and the KSA [14,15].

Diabates-related knowledge assessment tools play a crucial role in evaluating patients’ understanding of diabetes and its management. In the past, some authors attempted to evaluate diabetes knowledge among different types of diabetes patients using different international tools, such as the Michigan Diabetes Knowledge Test (MDKT), the Diabetes Knowledge TTest 2 (DKT 2), and self-developed tools [11,16,17,18]. Each tool has its advantages and limitations. For example, some tools may not assess all domains of diabetes knowledge and lack cultural relevance, particularly in addressing region-specific dietary practices. Hence, there is always a need for contextually appropriate and capable of identifying knowledge gaps specific to the target population.

Given the significant role of diet in diabetes management, this study focused on exploring nutritional knowledge and dietary patterns, as these are modifiable factors critical for glycemic control and reducing the risk of complications. Evaluation of dietary practices and their knowledge among patients with T2DM is important for several reasons. To begin with, dietary intake is important for controlling diabetes because it affects blood glucose concentrations, weight, and the possible development of complications [13,15,19]. Unhealthy eating patterns are not only contributing to the progression of T2DM but also predispose to other diseases such as hypertension and diseases of the cardiovascular system [20,21]. Secondly, knowledge about patients’ eating practices can help healthcare providers design appropriate education and intervention programs that are culturally relevant to their particular population. Lastly, such evaluations can be more revealing as they may also bring to light dietary management practices and problems that are wider in scope and thus would be useful to policymakers and health officials focusing on the diabetes pandemic [12,19]. Therefore, we require region-specific data to make appropriate and tailored nutritional education strategies for T2DM patients, which will prevent complications and decrease the burden on the healthcare system. Hence, continuous monitoring of these patterns and related factors is necessary. Given these considerations, this study aimed to evaluate the dietary knowledge and practices of T2DM patients in Central Saudi Arabia, providing insights to design culturally relevant and region-specific interventions.

## 2. Materials and Methods

### 2.1. Study Design and Setting

The present research was conducted from July 2024 to December 2024. The participants were recruited from the diabetes care center at King Fahd Specialist Hospital (KFSH) in Buraydah, Qassim region, which is situated in the central part of KSA. A total of 5696 T2DM patients were registered during the study period. We included all adult (18 years and above) T2DM patients with at least one follow-up visit. The participants included all genders, those who were mentally stable, and those willing to participate. The authors excluded all other forms of diabetes patients, hospitalized patients, and unwilling participants. The study design, sampling strategies, and recruitment process for T2DM patients from the current study’s setting are shown in Figure 1.

### 2.2. Sampling Procedures

We used an online sample size calculator to calculate the necessary number of T2DM patients to draw a valid conclusion from the research [22]. During the estimation, we used a 50% expected proportion and a 95% confidence level, a 5% margin of error, and a total population of 5696 registered T2DM, and it was concluded that a minimum of 360 participants were required for the current research. Using the convenience sampling method, we selected eligible T2DM patients. However, we followed some additional methods, including selecting only one T2DM patient from a family in the study, and we restricted a maximum of 10 patients in a day to recruit the participants over a period of time. The selection was based on the order of patient arrival at the clinic and availability for participation during their visit. No preference was given based on demographics or clinical characteristics, ensuring an unbiased approach within the convenience sampling framework.

### 2.3. Data Collection Methods

This research proposal was ethically approved by the Regional Research Ethics Committee, Qassim province (approval no: 607/45/429, dated: 17 July 2024), and we adhered to the guidelines of the Declaration of Helsinki. The T2DM patients were approached during the waiting time in the diabetes clinic. After giving brief information about the study and obtaining informed consent, we asked them to complete a pretested, validated, and standardized questionnaire. This study aimed to use a questionnaire tailored to the sociocultural and dietary context of KSA. Initially, the authors developed the questionnaire in English. However, the authors followed standard protocols in translating the questionnaire into Arabic (translation–back translation process) [23]. First, two translators proficient in both English and Arabic translated the entire questionnaire into Arabic. Subsequently, these translations were adjudicated by a panel of experts in the health sciences to obtain consistency in phrasing. After that, another blinded bilingual expert translated the original Arabic version back to English. The back-translated version was compared to the English original version to check whether there were disparities, and if so, reconciliation was performed. This back-and-forth process made it possible to obtain a final Arabic questionnaire that is linguistically and culturally relevant to the target group. The authors developed this questionnaire according to group discussion and brainstorming sessions that were based on the existing literature [8,24,25]. The content validity of the questionnaire was evaluated by a panel of five experts in diabetology, nutrition, and public health. The panel assessed the relevance and comprehensiveness of the items in relation to the study objectives. The developed tool was tested among 33 T2DM patients during the pilot study (face validity). All patients agreed that the knowledge and pattern assessment tool was simple and easy to answer.

The developed tool (please find more details in the Appendix A) consisted of three sections. The first section gathered the T2DM patients’ demographics and health-related attributes. The second (knowledge) section inquired about and assessed three different domains, including general knowledge about diabetes, including its causes, risk factors, and complications; understanding of blood glucose management, including target levels and the role of HbA1c; and knowledge about dietary practices, focusing on the impact of specific food groups on glycemic control. The third and final section assessed dietary patterns of T2DM through 10 questions on aspects such as sugary beverages, whole grains, processed foods, fatty foods, lean protein, sweets, healthy snacks, low-fat dairy, non-starchy vegetables, and low-sugar fruits in a typical week. In the knowledge section, the participants answered with “yes”, “no”, or “not sure”. Correct responses were awarded 1 point, while incorrect or not sure ones received 0 points. The correct response varied depending on the question, with some items having “Yes” as the correct answer and others having “No” as the correct answer. We made the cumulative knowledge score and divided them into low (<50% of total points), medium (50 to 75% of overall points), and high (>75% of overall points). In the dietary patterns section, the patients recorded their answers ranged from never to daily in a week (scores ranged from 1 to 5). Similarly to the knowledge, we computed the overall points in the dietary patterns and classified them into unhealthy dietary patterns (<34%), moderately healthy dietary patterns (34 to 67%), and healthy dietary patterns (>67%). The reliability of the questionnaire was assessed during the pilot study using Cronbach’s alpha. Regarding Cronbach’s alpha, we obtained 0.85 and 0.80 for the knowledge and dietary patterns in the pilot study, respectively. These values indicate acceptable internal consistency. Furthermore, test–retest reliability was conducted with a subset of 10 participants, with an interval of two weeks, yielding a correlation coefficient of 0.88, demonstrating strong reliability over time.

### 2.4. Data Analysis

The authors used the Statistical Package for the Social Sciences (SPSS, V 23.0) for current research data analysis and to apply appropriate statistical tests. The results of patients’ demographics, health-related characteristics, responses in knowledge, and dietary patterns are shown as frequency and proportion. Considering that the current study did not meet the normality assumption determined through the Shapiro–Wilk’s test, Spearman’s analysis evaluated the correlation between knowledge and dietary patterns. We applied multivariate analysis (binomial logistic regression) to assess the factors related to knowledge and dietary patterns. We combined low and medium knowledge for this model and compared it with high knowledge. Similarly, healthy dietary patterns were compared with moderate and unhealthy dietary patterns. A *p*-value of less than 0.05 was set as a statistically significant value.

## 3. Results

The authors contacted 409 eligible T2DM patients to obtain the desired sample size (363, response rate: 88.8%). These 409 patients were identified as eligible for the study based on the inclusion criteria. These patients were approached during their routine follow-up visits to the diabetes care center. Of the studied T2DM patients, most of them belonged to the age group of 45 to 60 years (40.2%), nearly half of them were males (50.1%), currently married (76.0%), working in government (36.6%) and private sectors (38.6%), 40.2% suffered with diabetes for 5 to 10 years, and 54.3% were treated with oral hypoglycemic agents (OHA) (Table 1).

Regarding participants’ responses in knowledge items, the statement “Diabetes can be completely cured” received the highest correct responses (79.9%), followed by fasting blood sugar level (69.1%) and intake of low-fat diet food and the lower risk of diabetes (66.4%). The lowest correct responses were observed in the statement related to the link between high-caloric food and blood sugar levels (30.6%) and the impact of soft drinks on blood sugar (49.9%) (Table 2).

Table 3 depicts the diabetes-related dietary pattern among the participated T2DM patients. Of the studied patients, 44.6% consumed a diet with whole grains daily, 37.2% preferred lean meat protein sources, 35.5% rarely, and 15.2% never consumed fried food items such as fried chicken and potato chips. Furthermore, we observed a positive correlation between diabetes-related knowledge and patterns among T2DM patients (rho = 0.627, *p* = 0.001).

Of the 363 T2DM patients, low, medium, and high diabetes-related knowledge was observed in 36.4%, 34.4%, and 29.2%, respectively. Regarding dietary patterns, unhealthy, moderately healthy, and healthy dietary patterns were noted in 34.4%, 33.3%, and 32.3%, respectively (Table 4).

Subgroup analysis of diabetes knowledge and pattern revealed that females had lower levels of knowledge than males (*p* = 0.013). However, there was no significant difference in the diabetes-related dietary patterns between males and females (Table 5).

The binomial regression findings of an association between diabetes-related knowledge and T2DM patients’ background characteristics are shown in Table 6. The knowledge level was significantly lower among females (adjusted OR [AOR] = 0.64, 95% CI = 0.49–0.81, *p* = 0.018). Significantly higher levels of knowledge were observed among the following variables: patients with a family history of diabetes (AOR = 2.73, 95% CI = 1.42–4.27, *p* = 0.007), duration of diabetes 5 to 10 years (AOR = 2.34, 95% CI = 1.18–3.66, *p* = 0.039), duration of diabetes more than 10 years (AOR = 3.30, 95% CI = 1.16–4.57, *p* = 0.001), and those who follow-up regularly (AOR = 2.97, *p* = 0.003).

The binomial regression findings of an association between diabetes-related dietary patterns and T2DM patients’ background characteristics are shown in Table 7. Significantly higher levels of healthy patterns were observed among the following variables: patients with the age group more than 60 years (AOR = 2.99, 95% CI = 1.55–4.80, *p* = 0.001), duration of diabetes more than 10 years (AOR = 2.72, 95% CI = 1.20–3.69, *p* = 0.032), and presence of other chronic diseases (AOR = 2.25, *p* = 0.028).

## 4. Discussion

The present study assessed diabetes-related knowledge, patterns, and associated factors among T2DM patients from Central KSA. The responses to knowledge-related items point to variable levels of awareness around some of the critical aspects of diabetes management among the participants enrolled. For example, a substantial number responded correctly that diabetes cannot be completely cured. This is indicative of accepting the belief that T2DM is a chronic disease and requires lifelong to be managed. In contrast, Dinesh et al. found that only about 53% agreed with this statement [26]. This could be due to the study setting, as the latter study was conducted in rural areas. Among the proportion who had such knowledge, however, very little was known concerning high-caloric food and high blood sugar levels. This critical observation was observed in several studies [27,28,29]. In fact, a study by Letta et al. stated that an alarming proportion of (97.8%) T2DM participants wrongly responded to the association between high-caloric food and diabetes-associated complications (heart disease) [27]. Correcting these deficiencies would allow patients and policymakers to make appropriate dietary decisions that would always lead to appropriate clinical benefits in terms of improved glycemic control and long-term prognosis by enhancing adherence to recommended dietary practices, reducing intake of high-glycemic foods, and promoting balanced nutrition.

A high level of knowledge is essential for the patients to adhere to treatment recommendations, including diet, exercise, and medications. However, we found nearly one-third of the participants had low levels of knowledge. Similarly to the present study, these concerning observations were noted by several authors in overall knowledge scores or individual domains [28,30,31,32]. Therefore, improving this knowledge gap is crucial to empower patients to take an active role in self-care, including dietary management and improving long-term outcomes. A significant gap in knowledge concerning diabetes was registered between men and women, with lower levels of knowledge significantly observed among female participants. These findings are consistent with other findings and differ from some research, which considers the gender of individuals a possible determinant in the provision and/or understanding of health information, literacy, or participation in education programs on diabetes. Variations across the studies from different settings could be due to differences in sociocultural factors, inclusion criteria of the participants, and the tools used to evaluate diabetes knowledge. For example, Alemayehu et al. stated that being male is significantly associated with higher knowledge [33]. In contrast, some studies found no association between gender and diabetes knowledge [9,16]. Moreover, patients with a positive family history of diabetes and a longer duration of T2DM showed a higher knowledge level. These findings indicate that experience through themselves or their relatives is associated with greater disease awareness, probably due to more doctor visits or learning experience accumulation with time. It is worth noting that poor knowledge was observed in hypertension-related knowledge also among Saudi hypertensive patients [17,34].

It is crucial to discuss how other studies have utilized established knowledge assessment tools, such as the MDKT, to evaluate diabetes-related knowledge and its outcomes in various populations. For example, Al-Nozha et al. assessed diabetes-related knowledge among T2DM and type 1 diabetes patients using the MDKT [18]. They did not reveal any significant association between evaluated sociodemographic factors. However, they found that T2DM patients had more knowledge than type 1 diabetes patients. Another study conducted using an author-developed and validated questionnaire revealed an inadequate knowledge of different aspects of diabetes among the study participants [11]. A study conducted using 14-item MDKT revealed adequate knowledge among participants [35]. In contrast, using the DKT 2 tool, Zowgar et al. stated that diabetes-related knowledge was poor among their study participants [16]. Similarly to our study, Zwane et al. in South Africa revealed significant gaps in diabetes-related knowledge and suboptimal dietary practices among patients. While 36% of participants in our study exhibited low knowledge levels, Zwane et al. stated that about 55% of South African patients had average diabetes knowledge [36]. Both studies emphasize the global challenge of improving diabetes knowledge through health education tailored to cultural and dietary contexts and suggest the need for region-specific interventions tailored to the unique needs of each population.

The dietary trends and habits of T2DM patients in this study present certain favorable patterns as well as deficiencies, which warrant targeting interventions. Specifically, those patients who responded favorably to diabetes-related diet knowledge had better scores on dietary patterns overall. This finding indicates the importance of educational orientation for changing eating habits. Such an alteration may then be sufficient to lead to lasting changes in dietary behavior by making the patients appreciate the connection between their food consumption and their blood glucose levels. Our findings and statements are supported by Abboud et al. and Adam et al. Their study found a positive association between these two domains [24,37]. Binomial regression analysis indicated that the dietary patterns of patients above the age of 60 years, those who had had diabetes for longer durations, and those with other chronic illnesses were likely to adopt healthy dietary patterns. This result may be because they have been exposed to various healthcare providers, diabetes education, and chronic condition self-management over a longer duration. Additionally, older patients and those with other chronic diseases may pay more attention to their diet, given their greater understanding of the implications of lacking glycemic control. A recent study by Aljahdali et al. reported that age was an important associated factor in all domains of dietary patterns [25]. In contrast to the present study, Abboud et al. showed a positive association of dietary patterns with occupation and income [24]. There are both positive and negative aspects in the dietary patterns in the evaluated domains. On the one hand, there seems to be a trend toward eating more healthy products, while on the other hand, a significant part of the population still consumes sugar-sweetened beverages and processed foods. This is dangerous, as diabetes is becoming more prevalent, and an even more frightening thought is that, for many, it remains unnoticed. A survey by Rashid et al. found that their study participants’ dietary patterns were suboptimal in several domains, such as vegetables and quantity of starchy carbohydrates [38]. Sociocultural factors mainly influence dietary patterns and their associated factors. Hence, there are significant variations across the studies regarding patterns (overall and in each domain) and related factors [39,40,41]. This study’s results are critical to clinical practice and public health policy. The observed gaps in diabetes-related knowledge suggest the urgency of such health education programs. Policymakers prioritize implementing structured educational interventions through the healthcare providers focused on improving their understanding of the disease. From the public health perspective, these results highlight the need to develop culturally tailored approaches to diabetes management in the context of Saudi Arabia. Region-specific dietary recommendations should be incorporated into public health initiatives, both with regionally relevant examples of healthy foods and to incentivize adherence. Healthcare providers must be encouraged to integrate routine assessment of patients’ knowledge and dietary habits. They can supplement this information and identify high-risk individuals who need more intense education and support through future practice. Also, policymakers should think about including these reports in national programs of diabetes prevention and administration, where educational assets and interventions must be available to all socioeconomic groups.

The authors insist that some limitations are to be noted while reading the study findings. Firstly, the generalizability of the current study’s findings is limited due to substantial sociocultural variations across KSA and other GCC countries. Secondly, the authors found an association between these two outcomes and specific characteristics of the T2DM patients, not the causation (due to inherent limitations of cross-sectional study). Next, the bias due to recall and exaggerated responses must be kept in mind by the readers while interpreting the results. Other notable limitations include the study not assessing participants who had received prior diabetes education from healthcare professionals or nutritionists at a nutrition clinic, not evaluating their glycemic control (HbA1C level) status, and not evaluating the presence of diabetes-related complications. Addressing these factors in future studies could offer a more comprehensive understanding of the relationship between knowledge, behavior, and clinical outcomes. Furthermore, this study assessed diabetes-related knowledge and dietary patterns, but other modifiable factors, such as weight loss, physical activity, and medication adherence, are crucial in T2DM management. The literature on insulin sensitivity, glycemic control, and reducing cardiovascular risk is well documented through achieving weight loss and regular physical activity. Future research could explore these dimensions to provide a more comprehensive understanding of the behavioral and clinical determinants of diabetes management outcomes.

## 5. Conclusions

We explored a substantial number of T2DM patients who had low to moderate knowledge related to the nutritional aspects of diabetes. The study also revealed one-third of participants were following a healthy dietary pattern. We also observed a positive correlation between dietary knowledge and healthier dietary practices. Therefore, improving patients’ knowledge would enhance healthy dietary patterns for better glycemic control and prevent diabetes-associated complications. The findings underscore the need for targeted health education strategies that address gaps in dietary knowledge and promote healthier eating behaviors among T2DM patients. Addressing these deficiencies could empower T2DM patients of central KSA to make informed dietary decisions. Finally, we recommend multicentric studies to find the regional variations across KSA.

## Figures and Tables

**Figure 1 healthcare-13-00233-f001:**
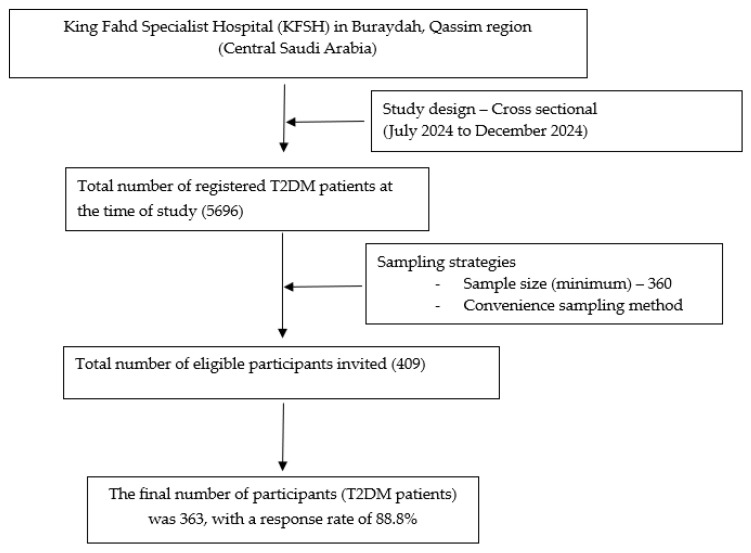
The study design, sampling strategies, and recruitment process for T2DM patients.

**Table 1 healthcare-13-00233-t001:** Demographic and diabetes-related characteristics of type 2 diabetes mellitus (T2DM) patients (n = 363).

Characteristics	Frequency (%)
Age group (years)	
Less than 45	116 (32.0)
45 to 60	146 (40.2)
More than 60	101 (27.8)
Gender	
Male	182 (50.1)
Female	181 (49.9)
Current marital status	
Married	276 (76.0)
Single	87 (24.0)
Level of education	
Primary	93 (25.6)
Secondary	146 (40.2)
Higher education (University)	124 (34.2)
Occupation	
Employee in government	133 (36.6)
Employee in private	140 (38.6)
Retired	47 (12.9)
Unemployed (including students, housewives)	43 (11.8)
Monthly income	
Less than 5000 SAR	76 (20.9)
5000–10,000 SAR	141 (38.8)
More than 10,000 SAR	146 (40.2)
Family history of diabetes	
No	133 (36.6)
Yes	230 (63.7)
Duration since diabetes diagnosis (years)	
Less than 5	90 (24.8)
5 to 10	146 (40.2)
More than 10	127 (35.0)
Current diabetes management method	
Oral hypoglycemic agents (OHA)	197 (54.3)
Insulin	130 (35.8)
Both insulin and OHA	36 (9.9)
Regular follow-up visits	
No	92 (25.3)
Yes	271 (74.7)
Associated with other chronic disease (s)	
No	219 (60.3)
Yes	144 (39.7)

**Table 2 healthcare-13-00233-t002:** Responses of T2DM patients to diabetes-related knowledge items (n = 363).

Knowledge Items	Correct Answer	Wrong Asnwer
	n (%)	n (%)
High sugar intake contributes to the development of diabetes.	237 (65.3)	126 (34.7)
If I have diabetes, my children are at a higher risk of developing it	189 (52.1)	174 (47.9)
Diabetes can be completely cured *	290 (79.9)	73 (20.1)
A fasting blood glucose level of 200 is high.	251 (69.1)	112 (30.9)
Diabetes can damage my kidneys.	236 (65.0)	127 (35.0)
Glycosylated hemoglobin (HbA1c) measures blood sugar values over the past week *.	206 (56.7)	157 (43.3)
Unsweetened fruit juice raises blood glucose levels	201 (55.4)	162 (44.6)
A can of diet soft drink is appropriate for treating low blood glucose levels *.	181 (49.9)	182 (50.1)
Eating foods with high calories increase blood glucose level	111 (30.6)	252 (69.4)
Eating foods lower in fat can decrease the risk of heart disease	241 (66.4)	122 (33.6)

* Negative questions (“no” is the correct answer).

**Table 3 healthcare-13-00233-t003:** Responses of T2DM patients to items assessing dietary patterns and eating habits (n = 363).

Pattern	Never n (%)	Rarely n (%)	Occasionally n (%)	Often 4–6 Times a Week n (%)	Daily n (%)
Frequency of consuming sugary beverages (e.g., soda, fruit juice, sweetened coffee/tea) per week	13 (3.6)	52 (14.6)	103 (28.4)	54 (14.9)	140 (38.6)
Frequency of whole grain consumption (e.g., whole wheat bread, brown rice) in the diet	41 (11.3)	45 (12.4)	57 (15.7)	58 (16.0)	162 (44.6)
Consumption of processed or fast foods (e.g., burgers, fries, pizza)	79 (21.8)	126 (34.7)	88 (24.2)	38 (10.5)	32 (8.8)
Frequency of consuming fatty or fried foods (e.g., fried chicken, potato chips)	55 (15.2)	129 (35.5)	110 (30.3)	39 (10.7)	30 (8.3)
Consumption of lean protein sources (e.g., poultry, fish, legumes).	25 (6.9)	55 (15.2)	76 (20.9)	72 (19.8)	135 (37.2)
Frequency of consuming sweets between meals	10 (2.8)	66 (18.2)	138 (38.0)	60 (16.5)	89 (24.5)
Frequency of consuming healthy snacks between meals (e.g., nuts, yogurt).	30 (8.3)	61 (16.8)	141 (38.8)	85 (23.4)	46 (12.7)
Frequency of consuming low-fat or fat-free dairy products.	40 (11.0)	41 (11.3)	80 (22.0)	119 (32.8)	83 (22.9)
Frequency of consuming non-starchy vegetables (e.g., leafy greens, broccoli).	37 (10.2)	84 (23.1)	122 (33.6)	73 (20.1)	47 (12.9)
Frequency of consuming low sugar fruits (e.g., apples, berries).	43 (11.8)	150 (41.3)	97 (26.7)	43 (11.8)	30 (8.3)

**Table 4 healthcare-13-00233-t004:** Diabetes-related knowledge and dietary pattern categories among T2DM patients (n = 363).

Variable	Frequency
Low (<50%)	132 (36.4)
Medium (51 to 75%)	125 (34.4)
High (>75%)	106 (29.2)
Unhealthy (<34%)	125 (34.4)
Moderately Healthy (34 to 67%)	121 (33.3)
Healthy (>67%)	117 (32.3)

**Table 5 healthcare-13-00233-t005:** Subgroup analysis of gender with the dietary knowledge and patterns of T2DM patients (n = 363). Test applied—Chi square test.

Variable	Total	Male n (%)	Female n (%)	*p*-Value
Low (<50%)	132	54 (40.9)	78 (59.1)	
Medium (51 to 75%)	125	74 (59.2)	51 (40.8)	0.013
High (>75%)	106	54 (50.9)	52 (49.1)	
Unhealthy (<34%)	125	61 (48.8)	64 (51.2)	
Moderately Healthy (34 to 67%)	121	63 (52.1)	58 (47.9)	0.867
Healthy (>67%)	117	58 (49.6)	59 (50.4)	

**Table 6 healthcare-13-00233-t006:** Binomial regression analysis of factors associated with diabetes-related knowledge levels among T2DM patients (n = 363).

Variables	Total n = 363	Knowledge			
		Low and Medium n = 257	High n = 106	Adjusted OR (95% CI)	*p*-Value
Age group (years)					
Less than 45	116	82	34	Reference (Ref)	
45 to 60	146	100	46	1.95 (0.83–3.70)	0.093
More than 60	101	75	26	1.57 (0.83–2.97)	0.164
Gender					
Male	182	124	58	Ref	
Female	181	133	48	0.64 (0.49–0.81)	0.018
Current marital status					
Married	276	196	80	Ref	
Single	87	61	26	1.04 (0.58–1.88)	0.886
Level of education					
Primary	93	68	25	Ref	
Secondary	146	104	42	0.95 (0.45–2.03)	0.909
University or above	124	85	39	0.96 (0.51–1.82)	0.908
Occupation					
Government	133	94	39	Ref	
Private	140	98	42	0.58 (0.16–2.11)	0.412
Retired	47	32	15	0.70 (0.20–2.43)	0.574
Unemployed	43	33	10	0.61 (0.16–2.36)	0.481
Monthly income					
Less than 5000 SAR	76	61	15	Ref	
5000–10,000 SAR	141	93	48	0.42 (0.15–1.21)	0.101
More than 10,000 SAR	146	103	43	1.36 (0.77–2.39)	0.285
Family history of diabetes					
No	133	99	34	Ref	
Yes	230	158	72	2.73 (1.42–4.27)	0.007
Duration since diabetes diagnosis (years)					
Less than 5	90	71	19	Ref	
5 to 10	146	106	40	2.34 (1.18–3.66)	0.039
More than 10	127	80	47	3.30 (1.16–4.57)	0.001
Current diabetes management method					
Oral hypoglycemic agents (OHA)	197	139	58	Ref	
Insulin	130	92	38	1.14 (0.47–2.81)	0.763
Both insulin and OHA	36	26	10	1.12 (0.46–2.76)	0.804
Regular follow-up visits					
No	92	71	21	Ref	
yes	271	186	85	2.97 (1.33–3.94)	0.003
Associated with other chronic disease (s)					
No	219	154	65	Ref	
Yes	144	103	41	1.10 (0.65–1.88)	0.717

**Table 7 healthcare-13-00233-t007:** Binomial regression analysis of factors associated with dietary patterns among T2DM patients (n = 363).

Variables	Total n = 363	Dietary Pattern			
		Moderate or Poor n = 246	Healthy n = 117	Adjusted OR (95% CI)	*p*-Value
Age group (years)					
Less than 45	116	92	24	Ref	
45 to 60	146	103	43	1.16 (0.64–2.12)	0.616
More than 60	101	51	50	2.99 (1.55–4.80)	0.001
Gender					
Male	182	129	53	Ref	
Female	181	117	64	0.78 (0.49–1.27)	0.324
Current marital status					
Married	276	187	89	Ref	
Single	87	59	28	1.17 (0.66–2.08)	0.592
Level of education					
Primary	93	65	28	Ref	
Secondary	146	99	47	1.01 (0.49–2.11)	0.976
University or above	124	82	42	0.99 (0.53–1.84)	0.977
Occupation					
Employee in government	133	92	41	Ref	
Employee in private	140	88	52	1.07 (0.32–3.63)	0.908
Retired	47	33	14	1.43 (0.44–4.64)	0.549
Unemployed	43	33	10	0.93 (0.26–3.39)	0.916
Monthly income					
Less than 5000 SAR	76	59	17	Ref	
5000–10,000 SAR	141	93	48	0.52 (0.20–1.36)	0.183
More than 10,000 SAR	146	94	52	1.19 (0.69–2.07)	0.528
Family history of diabetes					
No	133	92	41	Ref	
Yes	230	154	76	0.70 (0.41–1.21)	0.207
Duration since diabetes diagnosis (years)					
Less than 5	90	109	37	Ref	
5 to 10	146	91	36	2.30 (0.86–4.58)	0.101
More than 10	127	46	44	2.72 (1.20–3.69)	0.032
Current diabetes management method					
Oral hypoglycemic agents (OHA)	197	139	58	Ref	
Insulin	130	88	42	0.52 (0.23–1.20)	0.129
Both insulin and OHA	36	19	17	0.55 (0.24–1.26)	0.159
Regular follow-up visits					
No	92	67	25	Ref	
Yes	271	179	92	1.47 (0.81–2.88)	0.307
Associated with other chronic disease (s)					
No	219	155	64	Ref	
Yes	144	91	53	2.25 (1.54–3.11)	0.028

## Data Availability

The raw data supporting the conclusions of this article will be made available by the authors upon request.

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
