# Peer review of "Assessment of Diabetes-Related Knowledge and Dietary Patterns Among Type 2 Diabetes Mellitus Patients in Central Saudi Arabia: Insights for Tailored Health Education Strategies"

_healthcare, 2025, doi:10.3390/healthcare13030233_

Round 1

Reviewer 1 Report

Comments and Suggestions for Authors

I read with interest the study presented by Alsaidan and colleagues on assessing Diabetes-related knowledge and dietary patterns among type 2 diabetic patients in a central region in Saudi Arabia. In this study, the authors have surveyed 363 adult patients with type 2 diabetes from the clinic, and excluded those who were hospitalised or with other types of diabetes. 

The study highlights some associations between the degree of patients' knowledge of their case, i.e., diabetes, and their dietary habits. Such a study fits the aim and scope of the journal, and is undoubtedly of interest to the readership of the journal given the prevalence of type 2 diabetes, both globally and locally.

Upon carefully reviewing the manuscript, several major issues were raised and need to be carefully addressed by the authors:

1-  The distribution of paragraphs in the introduction is quite confusing. For example, in line 47, authors stated "However, globally and in the KSA, several studies observed critical gaps in knowledge in these aspects.". This is then followed by a description of 2 studies in Thailand and Portugal! 

2- The presentation of the Methods section, in its current form, is significantly deficient and would invalidate the findings of this study. The authors in this study came up with a new questionnaire and claimed in lines 113-114 that it was "pretested, validated, and standardized". However, it is entirely unclear how these processes were conducted. How were the questionnaire's validity and reliability ensured?

3- Following up on the previous comment, the establishment of a new study tool, i.e., a questionnaire based on three studies only, is deficient, to say the least. It is unclear why authors did not resort to using truly standardized tests for diabetes knowledge, like the Michigan Diabetes Knowledge Test (MDKT). Nevertheless, if the authors attempted to draft a new questionnaire, they should accurately specify the domains of knowledge that they attempted to assess. Otherwise, the study tool is no more than a pop quiz for diabetic patients.

4- It is suggested that the study tool be included in the manuscript as a supplementary file.

5- It is unclear what language the questionnaire was administered in. Authors should clearly describe it, and if it is in a language different than English, the translation process should be thoroughly explained. 

6- In the results section, the presentation of the data seems inconsistent. For example, In tables 1,2 and 4 authors used separate columns for frequency and percentages, whereas in table 3 they put both frequency and percentages next to each other, which is preferrable.

7- Figures/Tables should come after they referenced in the text, not the other way around, like in Table 4.

8- The discussion paragraphs are incredibly long and difficult to follow. Please consider shortening them.

9- I also was expecting to see, in the discussion section,  how the new study tool, i.e., questionnaire, fared against other published and validated knowledge tools (10.3390/healthcare12171708).  

10- The limitations stated in the discussion seemed rather generative and poorly discussed. Authors should adequately discuss the limitations of their study, citing relevant references.

11-  Furthermore, there are more critical and clinically relevant limitations that the authors missed, and two examples will be given for demonstration. Firstly, the patients characteristics, which the authors based their comparisons on, should be clearly justified and possibly influence the outcome of their knowledge and dietary habits. However, authors did not assess whether these patients have already been adequately educated on their condition, e.g., through a professional healthcare worker. Similarly, some patients might have better dietary habits (and possibly knowledge) due to their attendance at a nutritionist clinic. Such factors were not explored at all, nor indicated as a limitation in this study.

Secondly, the authors assessed patients' knowledge of the condition and their dietary habits. However, the only clinically relevant criterion that was included was the presence of another chronic disease. One would have expected such knowledge and dietary habits to be linked with their primary condition, i.e., diabetes. For example, do subjects with higher knowledge and healthy eating habits have better glycemic control or fewer diabetic complications than those with lower knowledge and less healthy eating habits? 

The study has scientific potential and could add some perspectives to the current literature on diabetes in Saudi Arabia. However, unless the comments above are thoroughly considered and appropriately addressed, I cannot recommend publishing this article in its current state. 

Reviewer 2 Report

Comments and Suggestions for Authors

Comments  uploaded

Reviewer 3 Report

Comments and Suggestions for Authors

Dear Authors,

Your manuscript deals with an actual topic but should be improved and adapted. I hope to give you some suggestions and indications for doing that.

abstract: please specify why, and for what you assessed the diabetes-related knowledge and patterns. (to gain insights for tailored health education strategies?). in the conclusions of the abstract: please delete the last sentence.

Throughout the manuscript what is specific to central Saudi Arabia in your findings? I would explain the latter and not launch throughout the manuscript that interventions "should be tailored" to KSA (an intervention always has to be adapted to its context.)

proof the citations throughout the manuscript.

Introduction:

lines 42-43: in what way does the prevalence widely differ across the region? explain why you decided to "only "explore the nutritional knowledge and pattern.

line 46: what do you mean by "suggested diet"? suggested by whom?

line 48: which gaps did the studies observe? possibly add a sentence on this.

lines 80-81: I suggest deleting the sentence on globalisation as it isn't clear what is meant.

2.1. study design and setting: how did you get access to the DM2 patients? Do you work there? please explain.  Maybe you could differentiate between inclusion and exclusion criteria? How did you define "unwilling participants"?

2.2. Although you report how you calculated the sample size it remains unclear how you applied the convenience sampling: did you include all till you had the right number?
describe in more detail how you selected "eligible"  patients. how did you choose the 10 patients/day? (criteria)
Have you all or some of you been involved in the selection process?

2.3. line109: instead of "ethically cleared" state simply "approved by"?

line 113: it remains unclear to me till the end of the manuscript how you validated and "standardized" the questionnaire. I give you some questions to possibly answer: Why did you decide to develop your questionnaire and not use another valideáted one? how did you validate your tool? did the pilot testing occur before? Were all items/questions self-assessed?

lines 127-128: how did you define "correct" and "incorrect"?

lines 134-135: what does this mean? only in 33, did you apply Cronbach's Alpha???

Results: How did you contact the 409 patient? Please specify as already suggested.

 Table 1 (applies as well to others): Try to improve the readability of titles and variables in all tables. (Characteristics)

table 2: how did you handle missing values (such as in item 3?)

 Table 4: variable: how did you assign participants to the variables/categories?

Table 5: Specify "Ref"

Discussion:

lines 213-15: not specifically in Central KSA but everywhere. Please focus (as suggested before) on why and how it is crucial for effective health education strategies....

lines 228-230: how and why or in what way?

line 261: further analysis of what?

and line 265: a cross-sectional study does not have a "trend"...

somewhere you could discuss that weight loss, PA and other interventions would be factors for improvement, not targeted by your study.

line 285-286: did you target this with the study? or did you obtain/report some results on that?

lines 289-290: I think healthcare providers should do that. or have you explained how you think policymakers should target to that?

line 292: "Diabetes pathophysiology" is the first time nominated here.

line 296: referring to the instrument: if this is right you should explain it before as suggested.

lines 297- 298: I do not see the uniqueness. be more modest.

line 301: delete this sentence as you never mentioned the qualitative component of your study.

Conclusions:

 line 309: knowledge (only) related to the nutritional aspects (diabetes)

lines 315-319: delete or change the last two sentences. Their content has to be supported by the findings of your study.

Reviewer 4 Report

Comments and Suggestions for Authors

 In this study, A.A. Alsaidan et al. examined diabetes-related knowledge and dietary patterns in 409 patients with type 2 diabetes from Central Saudi Arabia. The results are of particular interest, especially given that the prevalence of the disease is particularly high in this part of the world. The study was conducted with the use of standardized and validated questionnaires. The article is written very clearly. To make the results more meaningful, I would recommend some revision.

1. The clinical characteristics of the patients are very incomplete. What was the level of glycated hemoglobin A1c in the included patients? If the level of HbA1c was measured, how did patients with different levels of diabetes-related knowledge and different dietary habits differ in this indicator?

2. The study included both patients on oral antihyperglycemic drugs and insulin therapy. Please provide details on the treatment. It is strange that a very small proportion of patients received a combination of insulin and oral antidiabetic drugs, while a significant proportion of patients received insulin as a monotherapy. Please explain.

3. How did the level of knowledge and dietary patterns differ between subjects on insulin and on oral antihyperglycemic drugs?

4. Are there any standards for diabetes education in Saudi Arabia? Did the included patients previously undergo therapeutic training in diabetes management?

5. Table 2. What do the asterisks mean?

Round 2

Reviewer 1 Report

Comments and Suggestions for Authors

I congratulate the authors on their hard work addressing all the comments raised in the first round of revision.

I have no further comments, and I wish them the best of luck in conducting follow-up studies that build on the findings of this study, and address its limitations.

Author Response

Dear Reviewer,

We sincerely thank you for your kind words and encouragement. The authors believe that the reviewer's suggestions have significantly improved the quality of our manuscript. We deeply appreciate your time and effort in reviewing our work, and we are motivated to build on these findings in future studies while addressing the limitations identified. Thank you once again for your valuable support and best wishes.

Sincerely yours

Reviewer 3 Report

Comments and Suggestions for Authors

Dear Authors,

Thank you for the revised manuscript. The introduction chapter still needs a section about knowledge assessment tools. Specify which common tools (internationally) exist and why you did not use one of them. Please justify why there was a need to develop your own tool. 

Author Response

Dear reviewer,
Thank you once again for your valuable time spent reviewing our revised paper and for your positive feedback.
According to the reviewer's suggestions, we included a section about different tools used internationally with relevant references. In the revised manuscript, we have also justified the development of our own tool, emphasizing its cultural relevance and adaptation to the specific dietary habits and sociocultural context of T2DM patients.
Furthermore, in continuation, we added a section about comparing the studies using different tools in the discussion section.
Thank you once again for the positive and constructive feedback.
Looking forward to hearing from you soon.
Thanks and regards

Reviewer 4 Report

Comments and Suggestions for Authors

The article has been successfully revised in accordance with the reviewer's comments.

Author Response

(The authors gave the same response as above.)
